# Analysing the Impact of Harvesting Methods on the Quantity of Harvesting Residues: An Australian Case Study

**Mohammad Reza Ghaffariyan [1],\* and Eloïse Dupuis [2]**

[1] Forest Industries Research Centre, University of the Sunshine Coast, Locked Bag 4, Maroochydore DC, Sunshine Coast, QLD 4558, Australia

[2] Renewable Materials Research Centre, Department of Wood and Forest Sciences, Laval University, 2405 Terrasse Street, Quebec City, QC G1V 0A6, Canada; eloise.dupuis.1@ulaval.ca

\* Correspondence: mghaffar@usc.edu.au

**Abstract:** Many parameters can influence the weight of harvesting residues per hectare that remain on plantation sites after extracting sawlogs and pulpwoods. This study aimed at quantifying the impact of the cut-to-length and whole-tree harvesting methods on the weight of harvesting residues using 26 case studies in Australian plantations. A database was created using case studies conducted in different plantations, to measure the weight of harvesting residues per hectare and the components of harvesting residues. An analysis of variance was applied to test the impact made by the harvesting methods. The results confirmed that the cut-to-length harvesting method produced a larger weight of residues (104.0 tonnes of wet matter per hectare (tWM/ha) without additional biomass recovery and 64.7 tWM/ha with additional biomass recovery after sawlog/pulpwood extraction) than the whole-tree harvesting method (12.5 tWM/ha). The fraction test showed that stem wood formed the largest proportion of the harvesting residues in cut-to-length sites and needles were the largest component of the pine harvesting residues in sites cleared by the whole-tree harvesting method. The outcomes of this study could assist plantation managers to set proper strategies for harvesting residues management. Future research could study the impact of product type, silvicultural regime, stand quality, age, equipment, etc., on the weight of harvesting residues.

**Keywords:** harvesting method; residues; stem wood; branches; needles





## 1. Introduction

The weight of harvesting residues on sites after commercial wood recovery depends on various parameters such as applied harvesting method, equipment, product type, silvicultural regime, species, site, stand age, diameter at breast height (DBH), and stand quality. Different terms such as forest residues, downed wood and deadwood are used to describe woody debris. Woody debris include fine woody materials (with a diameter less than 10 cm) and coarse woody materials (with a diameter larger than 10 cm) [1]. Harvesting residues herein refers to leaves and twigs (needles), cones, barks and branches with diameter larger than 3 cm were recorded within each sample [2,3]. Along with the weight of harvesting residues, the applied harvesting method can influence the type and quantity of the remaining harvesting residues. The cut-to-length method (CTL) is a harvesting method that processes trees into short or long logs at the stump to extract sawlogs and pulpwoods to the roadside, while fine woody materials and coarse woody materials are left on the operation sites. By using the whole-tree harvesting method (WTH), the harvesting residues are removed from the site to be concentrated at the landings [4,5]. With this method, some fine woody materials (with a diameter less than 7 cm) are inevitably left on the site (mostly needles/leaves and twigs) while a majority of the coarse woody materials (with a diameter larger than 7 cm) is recovered [6]. In Australian pine plantations, previous case studies indicated that the cut-to-length harvesting method produced a significantly higher weight of residues than whole-tree harvesting: 77.6 tonnes of wet

matter per hectare (tWM/ha) and 31.7 tWM/ha, respectively [7]. In eucalyptus plantations, 106.5 tWM/ha of residues were left on the site after application of the CTL harvesting method, but only 6.1 tWM/ha was left after applying the WTR harvesting method [3]. A case study in Norway spruce indicated that the weights of harvesting residues per hectare varied from 19.4 to 49.9 tWM/ha in the stand harvested by the CTL harvesting method after biomass recovery with a slash bundler [8]. When southern pine forests were clear felled, the quantity of harvesting residues was 44.7 tonnes of dry matter per hectare (tDM/ha) [9]. Another study reported by Watson et al. (1986) [10] indicated that 75.4 tWM/ha of residues were left on a clear-felled slash pine plantation, whereas the average weight of harvesting residues for mixed slash and loblolly pine stands was 61.5 tWM/ha after applying the CTL harvesting method.

From a plantation management perspective, the most common reasons to collect harvesting residues from sites include reducing the fire hazard by removing fuel from the forest floor, minimizing the beetle attack hazard and preparing the site for tree planting [11]. Studies in the Southeast USA have indicated that there is an ecological relationship between the quantity of dead woody materials and wildlife which can differ depending on the forest types [12]. Coarse woody materials can also play a role to provide a suitable feedstock for bioenergy production as it does not compete with other commercial wood products (e.g., sawlog and pulpwood) for conventional timber utilizations [13]. Recovering harvesting residues can also increase growers' revenues [6]. From a bioenergy production perspective, it is possible to recover biomass after or along with the commercial timber harvesting, using an integrated biomass approach that allows producing two types of products (sawlogs/pulpwoods and biomass) [6]. However, accessible forest harvesting residues are not always easily recoverable. Thiffault et al. (2015) [13] reviewed the recovery rates of harvested biomass in over 60 studies conducted in boreal forests. The average biomass recovery rate in boreal forests was 52.2% [13].

Forest harvesting residues left on the forest floor can also provide several ecological advantages. They can improve the soil structure, help to resist the soil compaction, provide a buffer against erosion and improve water filtration [11]. On the other hand, Kuiper and Oldenburger (2006) [14] reported that studies conducted in many European countries have demonstrated that the crown mass removal might endanger the sustainability of the stands, depending on site characteristics, composition and the amount of recovered biomass. Intensive biomass harvesting, or crown mass removal may result in reduced soil fertility because most nutrient-rich components (e.g., leaves/needles and small woods with diameter less than 7 cm) are included in the crowns [1,15,16]. The removal of harvesting residues should be avoided in poor quality soils to help maintain nutrients [17,18]. Intensive biomass recovery can reduce soil fertility, which can negatively influence the stand growth over the longer term [19]. Several countries have established guidelines to limit nutrient removals especially due to removal of leaves and fine materials [1,20,21]. In Finland, it is recommended to leave at least 30% of the woody debris on sites [1,15], whereas in France, forest growers are recommended to leave at least 10% of fine materials (with a diameter less than 7 cm) in non-sensitive soils, and at least 30% in moderately sensitive soils [1]. Guidelines in the Pacific Northwest of the USA recommend leaving at least 4–5% of total above-ground biomass on sites to ensure that sustainability of the soils and growth of the stands are maintained [20].

This study aimed at estimating the weight of the post-harvesting residues per hectare and to determine the share of each biomass component using a larger data set compared with the previous studies [2]. The other objective was to determine the impact of harvesting methods on the weight of the remaining harvesting residues per hectare.

## 2. Materials and Methods

### 2.1. Study Area

Twenty-six case study areas located in a range of plantations were assessed within this study. The study areas included 4 sites in Tasmania (TAS), 2 sites in New South Wales

(NSW), 5 sites in South Australia (SA), 9 sites in Western Australia (WA) and 6 sites in Queensland (QLD). Ten new case studies were added to the previous database [2], and were located in Queensland (6 new case studies) and Western Australia (4 new case studies). The harvesting methods included the cut-to-length and whole-tree harvesting methods, with some sites treated for a secondary recovery of biomass for bioenergy purposes. The sites harvested by the CTL method were classified into two machines (harvester and forwarder: trees were felled and processed into logs by a harvester-processor then logs were extracted to the roadside by a forwarder) or three machines (feller-buncher, processor and forwarder: trees were first felled and bunched by a feller-buncher, then a processor processed the trees into short logs at the stump to be extracted to the roadside by a forwarder). On one of the sites harvested by the CTL harvesting method, a fuel-adapted harvesting method was applied, which consisted of piling the logs and the harvesting residues into distinct piles adjacent to the harvester's travel path [22]. WTH operations included two options: a combination of a feller-buncher, two grapple skidders and an in-field chipper at the roadside to chip whole trees or a combination of a feller-buncher, two grapple skidders and a processor processing trees into short logs at the roadside. The characteristics of the study sites are summarized in Table 1.

**Table 1.** Characteristics of the study sites.

|  | Cut-to-Length | Whole-Tree Harvesting |
|---|---|---|
| **Number of Case Studies** | **21** | **5** |
| Species | *Eucalyptus globulus* Labill, *Eucalyptus nitens* H. Deane&Meaden, *Pinus radiata* D.Don, *Pinus elliottii* Engelm, *P.elliottii x P.caribaea* Morelet *var.* | *Eucalyptus globulus* Labill, *Pinus elliottii* Engelm |
| Soil | Brown, Clay Loam, Sandy-clay, Yellow Earth | Brown, Sandy-clay, Yellow Earth |
| Product type | Pulpwoods, Sawlogs, Wood chips, Biomass residues | Pulpwoods, Sawlogs, Wood chips |
| Yield (tWM/ha) | 135–526 | 151–293 |
| Stand age (year) | 10–34 | 15–30 |
| DBH (cm) | 18–56 | 18–32 |
| Tree height (m) | 17–37 | 17–27 |
| SED * (cm) | 5–10 | 5–10 |

* SED: Small-end diameter used by the harvester.

### 2.2. Study Method

#### 2.2.1. Method for Estimating the Weight of Harvesting Residues per Hectare

A method that was developed by the Cooperative Research Centre (CRC) for Forestry in Australia for sampling harvesting residues [2,3], was applied to measure the weight of the harvesting residues and to estimate the percentage of each component including needles/leaves, barks, branches, stem woods and cones. This method for harvesting residues assessment is based on a study area of 1 ha and the sample area is 0.5 m × 0.5 m. After a brief investigation of each study site, two or three visually identifiable strata were defined. A set of representative transects was created at each study site to provide about 50 sample points (5 transects including 10 samples per each transect where spacing among transects was 20 m and where spacing among the samples on each transect was 10 m). The samples were visually assessed for which stratum the point represented, to produce a map of the stratum points (marked on the ground so they could easily be found for detailed sampling) and to determine the proportion of each stratum. Based on the transects, the number of points identified per stratum was entered into a spreadsheet to determine the number of pre-samples required per stratum (total numbers of pre-samples were 9–11).

Using the transect points, the required 0.5 m × 0.5 m grid pre-samples were randomly collected within each stratum. The plot samples were used to compute the number of samples per stratum. Absolute error was adjusted so that number of required plots was about 20 per hectare. If the error as a percentage of the mean was less than 15% using the transect points, additional 0.5 m × 0.5 m grid samples were randomly collected within each stratum, while ensuring pre-sample points were not repeated. For the fraction test per hectare, 5 samples were randomly selected in some of the study sites. The (green) weight of leaves and twigs (needles), cones, barks, and branches with diameter larger than 3 cm were recorded within each sample. Estimated weights of residues (tWM/ha) were then calculated by multiplying the mean weight of harvesting residues for the plots within each site by 10 to convert from $kg/m^2$ to tWM/ha. Moisture content was not measured during most of the case studies, thus tWM was used instead of tonnes of dry matter.

### 2.2.2. Statistical Analysis

One-way analysis of variance (ANOVA) was applied to test if there was any significant difference among the weights of harvesting residues under different harvesting methods. The normality of residuals and the homogeneity of variances were tested beforehand. Tukey's Honest Significant Difference (Tukey HSD) test was applied to identify which means were significantly different from others among the study treatments. The null hypothesis could be expressed as follows:

$H_0$: *Average weight of harvesting residues per hectare produced by the CTL method = Average weight of harvesting residues per hectare produced by the WTH method = Average weight of harvesting residues per hectare produced by CTL with biomass recovery.*

## 3. Results

### 3.1. Harvesting Residues

According to the results, the estimated weight of remaining harvesting residues varied highly among sites after applying various harvesting operation alternatives (Figure 1). In a case study in Southern Tasmania Pine plantation, 238.6 tWM/ha of remaining harvesting residues were left after using the CTL harvesting method, which was an exceptionally high level of harvesting residues in comparison with other pine plantations sites. The minimum weight of harvesting residues was 4.2 tWM/ha, which was observed on a Eucalypt plantation cleared by the WTR harvesting method, in the Southwest Australia.

Figure 2 provides the average weight of harvesting residues per hectare for three harvesting methods. The analysis of variance confirmed that there was a significant difference among the mean values of the weights of harvesting residues per hectare for the study treatments at $\alpha = 0.05$ (Table 2). This showed that the harvesting methods significantly impacted the weight of harvesting residues.

The Tukey HSD multiple comparison test confirmed that all three harvesting methods were different at $\alpha = 0.05$. Application of the whole-tree harvesting method left significantly lower weights of harvesting residues (average of 12.5 tWM/ha) than cut-to-length (104.0 tMW/ha) and cut-to-length method followed by the biomass recovery (64.7 tWM/ha). The statistical test confirmed that subsequent recovery of biomass after CTL harvesting resulted in significant lower weights of harvesting residues than the sites that were clear-felled with CTL without biomass recovery.

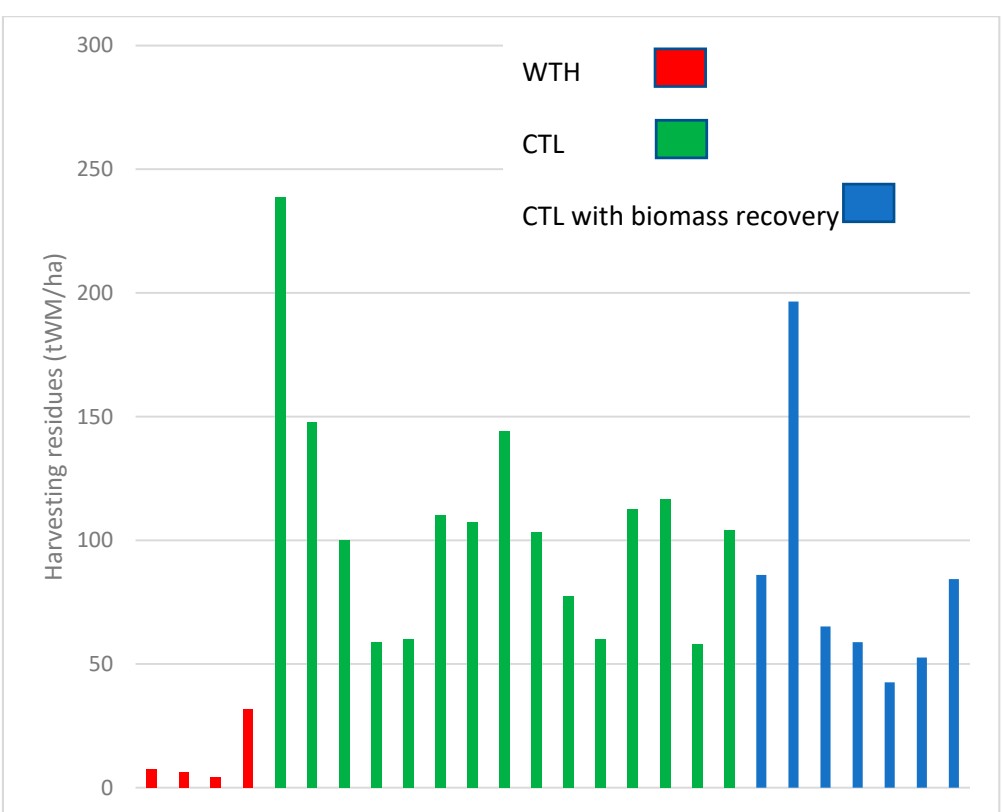

**Figure 1.** Weight of harvesting residues per hectare for all study sites and different harvesting methods.

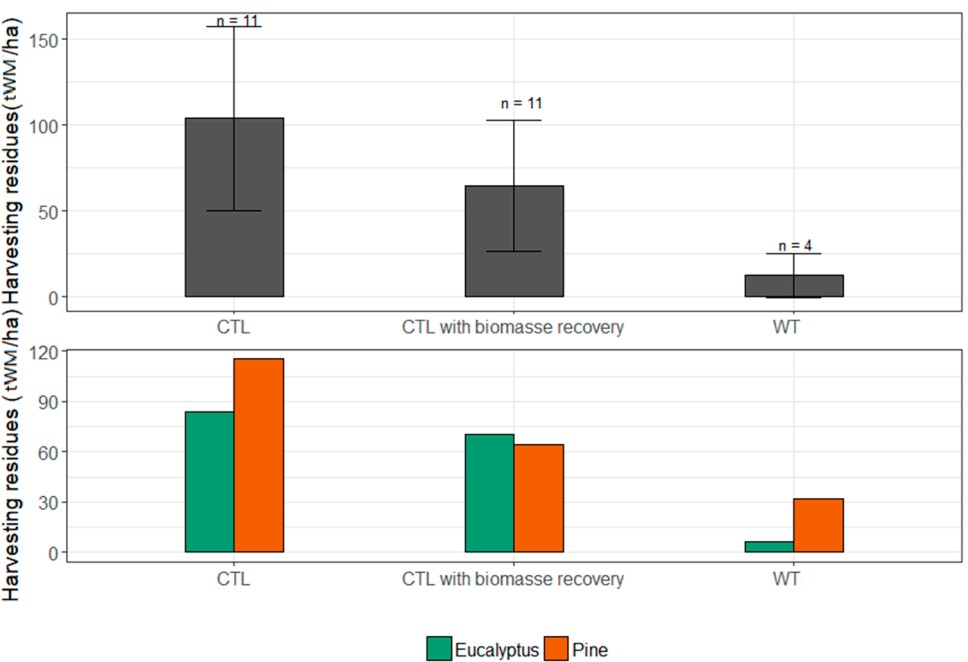

**Figure 2.** Average weight of harvesting residues per hectare for different harvesting methods and species. Error bars show standard deviation.

**Table 2.** Analysis of variance to test the impact of harvesting methods.

|  | Df | Sum of Squares | Mean Squares | F Value | Sig. Level |
|---|---|---|---|---|---|
| Factors | 2 | 25,685 | 12,842.60 | 5.93 | 0.00 |
| Residuals | 23 | 49,766 | 2163.80 |  |  |

The data analysis showed that the average weight of pine harvesting residues per hectare was higher than that of eucalypt harvesting residues for the CTL and WTH harvesting methods. When biomass recovery was applied following CTL, the weight of eucalypt harvesting residues per hectare was larger than that of pine harvesting residues, however, this occurred only on one case study area. Thus, further information is required to provide a more accurate estimate, which should be collected in future studies (Figure 2). It should be noted that the number of observations for the WTH method was not the same as for the CTL method, because there were less research trials conducted on WTH that collected information on the harvesting residues within our research program. Based on the analysis of variance (Table 3), species did not have any significant impact on the weight of harvesting residues at $\alpha = 0.05$ in this study.

**Table 3.** Analysis of variance to test the impact of species.

|  | Df | Sum of Squares | Mean Squares | F Value | Sig. Level |
|---|---|---|---|---|---|
| Factors | 1 | 3373.2 | 3373 | 1.23 | 0.29 |
| Residuals | 24 | 72,078 | 3003.3 |  |  |

*3.2. Biomass Recovery*

Biomass recovery rates varied depending on the harvesting equipment [2] (Table 4). Recovery rate for Pinox slash bundler in Eucalypt stands was 64.5%. When the harvesting residues were concentrated by an excavator to the rakes, it helped with increasing the recovery rate of slash bundler. If a slash bundler is applied to collect scattered harvesting residues on the site, its recovery rate might be lower [23]. A Bruks mobile chipper yielded a recovery rate ranging from 15.2% to 55% [3]. When the Bruks chipper was applied to collect only the residual stem woods, the recovery rate was low (15.2%); however, when it was used to collect all residues including stem woods and branches, the recovery rate increased to 55%. Conventional forwarders resulted in a range of recovery rates varying from 13.7% [24] to 42% [25], when recovering harvesting residues in pine stands. Forwarder's recovery capacity can increase as the weight of harvesting residues per hectare increases. In the areas with high weights of harvesting residues per hectare [25], the forwarder could work more efficiently and collected more biomass per hectare. Strandgard and Mitchell (2019) [22] reported a recovery rate of 68% for a conventional forwarder, when working under the fuel-adapted harvesting method.

**Table 4.** Biomass recovery rates for different harvesting technologies.

| Machine | Stand | Harvesting Method Prior to Biomass Recovery | Number of Cases | Recovery Rate (%) |
|---|---|---|---|---|
| Pinox slash bundler | Eucalypt | CTL | 1 | 64.5 |
| Bruks mobile chipper | Pine | CTL | 5 | 15.2–55 |
| Conventional forwarder | Pine | CTL | 2 | 13.7–42 |
| Conventional forwarder under fuel-adapted method | Pine | CTL | 1 | 68% |

### 3.3. Fraction Test of the Harvesting Residues

Within 16 case studies (15 pine case studies and 1 eucalypt case study), the share of each component of the harvesting residues was determined by separating and weighing each component during the field work (fraction test). As illustrated in Figure 3, the most substantial proportion of harvesting residues yielded by the CTL harvesting method and CTL followed by a biomass recovery belonged to the stem woods, branches and needles, while there was no stem wood among harvesting residues left after the WTH method. An important observation is that the weight of the needles represented over 50% of the weight of the remaining harvesting residues in the pine stands harvested by the WTR harvesting method. When the CTL harvesting method was applied in the pine stands, the stem wood represented the largest share of the harvesting residues. When biomass recovery occurred following CTL harvesting, the share of stem wood was slightly reduced because some of the stems were collected and removed from the stands for bioenergy usage. The smallest share of biomass in pine stands belonged to cones and barks (Figure 3).

For the eucalypt stands, only one case study within the data set was assessed for the fraction test (Figure 3), where the CTL harvesting method was applied to harvest the plantation area. The most substantial proportion of the harvesting residues was for the leaves, followed by branches and barks. Since the information was limited to only one case study, more data on Eucalypt stands are required in the future to get a more realistic estimation of biomass components (Figure 3).

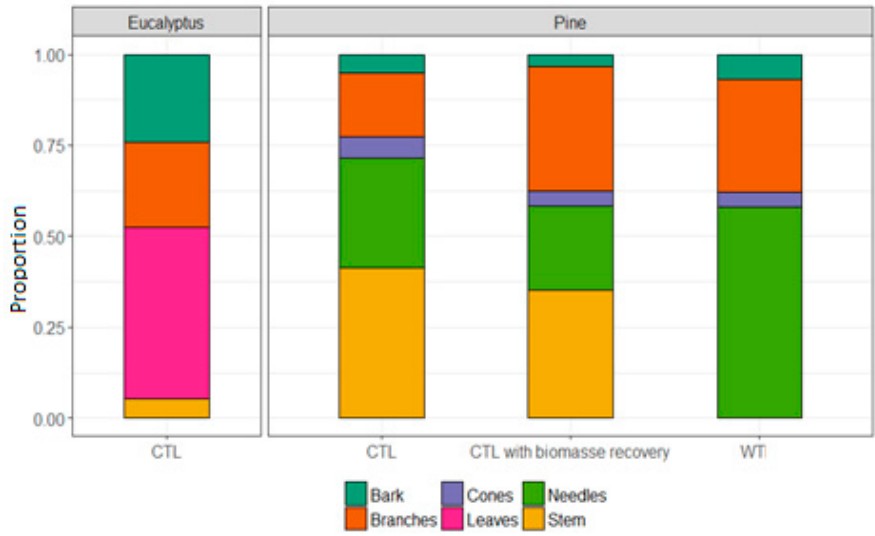

**Figure 3.** Relative proportion of biomass components.

### 4. Discussion

There was a high variability among the average weights per hectare of harvesting residues at different sites of this study (Figure 1). Some of the case studies within this project showed large weights of harvesting residues per hectare when applying the CTL harvesting method. This might be due to a higher yield per hectare, a higher stand quality, a small end diameter, or other factors, when applying the CTL harvesting method. Two case study areas with the highest weights of harvesting residues (without biomass recovery operation) were the sites with a largest small end diameter of 10 cm. Usually, after applying the CTL harvesting method, a large quantity of residues is produced in plantations, which was demonstrated by our study results. Other studies reported that the weight of pine harvesting residues per hectare following CTL can be near 160 tWM/ha [26,27]. The largest weight of pine harvesting residues in our study was 238.6 tWM/ha, which is slightly lower than the spruce case study in France (259.6 tWM/ha) reported by Cuchet et al. (2004) [28]. Our estimated weight of harvesting residues per hectare is also within the range reported by another study conducted by Bessaad et al. (2021) [1] in Central France, where the total

above ground biomass varied from 118 tWM/ha to 519 tWM/ha. For the stands managed under coppice regime, the weights of harvesting residue ranged from 174 tWM/ha to 230 tWM/ha [1]. In Eucalypt plantations, the average weight of harvesting residues per hectare in our study was slightly lower than in previous case studies [28,29], which could be impacted by higher yield and longer age in our case. A large weight of harvesting residues per hectare can be a suitable source for further utilizations (e.g., for bioenergy and biochar production) [30,31].

Crown removal in the WTH method resulted in lower weights of harvesting residues per hectares (Figure 2). Obviously, as study results confirmed, when residual stem woods and branches were recovered by special biomass collection machines (e.g., slash bundler or mobile chipper) the weight of remaining harvesting residues per hectare decreased, due to the partial removal of total weight of harvesting residues after CTL harvesting operation. The estimated recovery rates in this study (varying from 15.2% to 68%) is slightly consistent with the average biomass recovery rate of 36% reported by Thiffault et al. (2015) in non-Nordic countries [13]. Although slash bundler, mobile chipper and forwarder were considered in our study for collecting large proportion of residual stems and branches, another alternative could be recovering whole tops of trees, including branches [32], with forwarders. A dedicated compressing bunk can also be applied on the forwarders to increase the work efficiency and recovery rate [33]. Applying fuel-adapted technique [22] increased the recovery rates gained by a conventional forwarder. The average biomass recovery rate in pine stands was 41.6% for all types of biomass recovery machines.

Based on the fraction test results in pine stands, the needles, stems and branches represented the largest components for the three harvesting methods. This is similar to results from a recent study in pine plantations in New South Wales, Australia [31]. The fraction test results in pine stands harvested by the CTL harvesting method are consistent with a previous study finding in Swedish Norway Spruce forests, where the composition was 30–50% stem, 10–20% bark, 5–15% branches, 20–30% needles and 5–10% fine materials [34]. Another study indicated that 19% of the fresh harvesting residues in Norway Spruce belonged to needles [8]. High share of needles may positively contribute to maintaining nutrients on soils, as needles have the highest nutrient concentration in comparison with the other biomass components in pine stands, according to Ouro et al. (2001) [35] and Smethurst and Nambiar (1990) [36]. Needles also contain the highest decomposition and nutrient release rates [35,36]. Bessaad et al. (2020) [37] recommended that leaves/needles could be let fallen on the ground, before skidding the whole trees to the roadside/landing for returning nutrients to the soil and also for leaving easily degradable materials to assist the soil biological activity. Wall and Hytönen (2011) [20] identified needles as a significant source of nutrients for assisting the growth of the stands and to help mitigating the impact caused by the nutritional removal due to recovering some proportions of the harvesting residues. Past studies have indicated that intensive biomass recovery can result in loss of soil fertility and productivity [38–40], changing forest structure and making negative impacts on biodiversity [41], especially for species naturally adapted to sun-exposed conditions [42]. Water quality and ecosystem services may also be negatively impacted by intensive biomass recovery [42,43].

## 5. Conclusions

This study provided information on the quantity and composition of harvesting residues using several case studies conducted in Australian pine and eucalypt plantations. From the study results it can be concluded that the applied harvesting method can significantly impact the weight of harvesting residues per hectare. The CTL harvesting method produced larger weights of harvesting residues per hectare than the WTH method. Moreover, biomass recovery after CTL harvesting, carried out by any type of machines tested in this study, significantly reduced the weight of harvesting residues per hectare which were left on the site. Stem woods, needles and branches formed the largest components of the pine harvesting residues in sites logged by the CTL method. In pine

stands cleared by the WTH method, the needles represented over 50% of the harvesting residues, one of the largest fractions. Even though the weight of harvesting residues per hectare was low for sites harvested by the WTH method, the large share of needles could be useful for providing required nutrients (e.g., nitrogen) to the forest soils. However, more ecological studies are required to quantify the nutrient balance after applying the WTH method [44,45]. The impact of harvesting methods on the residues that were studied in this project are a significant outcome for the forest industry and academics in Australia and other regions. It can assist forest managers and harvesting planners to understand the impact of each harvesting methods in order to plan operations to sustainably maximize the value recovery, while controlling the potential nutrient removal and managing the soil fertility depending on the quantity of left residues and site conditions.

Future studies should verify the nutrient contents of the sites harvested by various timber harvesting methods to clearly identify which sites require further nutrients (e.g., fertilizers). A minimum required weight of harvesting residues should be determined in different stands and management conditions by future research [46], to move towards a more sustainable forest biomass utilization and coarse woody debris management [11,47,48] in Australia. More case studies will be required to verify the impact of some factors such as species, equipment, small end diameter, site, stand age and diameter at breast height (DBH) on the weight of harvesting residues per hectare. This will extend the pool of the database to accurately examine the impact of mentioned variables. The impact of product type, silvicultural regime, and stand quality could also influence the weight of harvesting residues per hectare, but these were not tested in this research. Understanding the effect of these parameters on the weight of harvesting residues could be a subject for future research. Moisture content of harvesting residues can also be measured in future trials to provide a more accurate estimate of the energy content of harvesting residues [49]. The capability of harvester's processing head data could also be considered by future studies to predict and control the level of harvesting residues [50].

**Author Contributions:** Conceptualization, M.R.G. and E.D.; methodology, M.R.G.; software, E.D.; validation, E.D. and M.R.G.; formal analysis, E.D. and M.R.G.; investigation, M.R.G.; resources, M.R.G.; data curation, E.D. and M.R.G.; writing—original draft preparation, E.D. and M.R.G.; writing—review and editing, M.R.G.; visualization, E.D.; supervision, M.R.G.; project administration, M.R.G.; funding acquisition, M.R.G. All authors have read and agreed to the published version of the manuscript.

**Funding:** This research received no external funding.

**Acknowledgments:** The authors would like to thank the following people who contributed to fieldwork measurements: Mauricio Acuna, Rick Mitchell, Martin Strandgard, Mark Brown, John Wiedemann, Damian Walsh, Michael Berry and Sam Van Holsbeeck. The authors also thank the Australian forest industry partners for providing information and access to their plantations and resources. We also thank three anonymous reviewers who provided valuable comments to improve the quality of the paper.

**Conflicts of Interest:** The authors declare no conflict of interest.

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
