# Peer review of "Analysing the Impact of Harvesting Methods on the Quantity of Harvesting Residues: An Australian Case Study"

_forests, doi:10.3390/f12091212_

Round 1

Reviewer 1 Report

The work deserves its publication, but it needs to be improved, both in formal aspects and in the depth of results analysis and discussion.

Regarding to formal aspects, I think that the text requires standardization in the use of initials, which must be explained in the first occasion where they appear. For example, the authors use GMT, I guess for Green Matter Tonnes, it should have been explained. Moreover, I think that there are innitials more commonly used, for example tWM (tonnes of wet matter) or, as the authors use in line 123, the most correct Mg - where the authors should explain that it refers to wet or green matter.

In general, I think that is much more standard to use tDM (tonnes of dry matter) as a unit more useful for comparison purposes, so I encourage the authors to use this unit to express their results in a more significant way.

On the other hand, the initials used for one of the compared harvesting systems (WTR) is not common, it is much more used Whole Tree Harvesting/ System (WTH-WTS), Full Tree Harvesting/System (FTH-FTS), I think that the text would be more understandable if the authors use one of these more usual abbreviated terms.   

In lines 109, 115 and 119 there are unnecesary repetitions, and there are three parentheses that do not close. In line 118, the phrase "(i.e. each 5th stratum 1 do plot)" is not clear, the authors should clarify its meaning.

In the Results Chapter, I do not understand the order of the study sites in Figure 1, it would me more clear if the harvesting systems were grouped (for example, first the block corresponding to WTH, then CTL, then CTL with biomass recovery).

As I have already mentioned, the results would me more significant if expressed ain terms of tonnes of dry matter, netter than tWM.

The table 1 should be Table 2.

The lack of influence of stand age and DBH which is described in lines 207-211 is a bit strange to me. I would suggest the authors to try the ANOVA separating species and/or harvesting systems, when the number of data were enough, I suspect that there should be significant differences for the same species and harvesting system if you compare DBH or age intervals.

Table 2 should be Table 3.

The biomass recovery analysis could be deepen, it would improve and provide more useful information if expressed as a Table. 

Last, but not least, the discussion is poor. The authors must revise other references to discuss their findings, as there are many references dealing with harvesting residues recovery rates, composition of harvesting residues, etc.

As the idea of revise several Australian studies about residues weigth, recovery rates , residues composition, etc. is actually interesting, it deserves publication, but in my opinion the authors must deepen their analysis of results and discussion, besides improving the indicated formal aspects.   

Author Response

1- GMt has been changed to tWM in whole text of paper.

2- Dry mass tonnes: This is a good suggestion however most of our case studies were limited by resources thus moisture content was not measured thus we presented the results in green tonnes. I have suggested future research to include MC and dry tonnes.

3- Name of harvesting methods were all changed as this reviewer suggested. This was done in whole text of the paper.

4- Repeated words were deleted as this reviewer asked (in page 3, description of sampling method)

5- Figure 1 was revised to reflect reviewer's suggestion on sorting the harvesting methods in a harmonised way.

6- Table 1 was changed to table 2.

7- The ANOVA and correlation analysis was performed as reviewer suggested to test the impact of DBH and species however it did not give any significant impact. Thus we removed this section as this is mainly caused by low number of observations for each class (page 5). In conclusions (page 7) I added further explanations to address this issue.

8- Table 2 was changed to Table 3 in the manuscript.

9- Biomass recovery was added into a new table. More explanations  on previous studies were added in page 6.

10- Lots of references were added to the discussion and introduction to address reviewer comment on enriching discussions.

Reviewer 2 Report

Duplicated publication !

https://research.usc.edu.au/view/delivery/61USC_INST/12131539930002621/13131539920002621

Author Response

I have explained to Quinn Zhang in an email dated 4/08/2021. This reference is only a short industry bulletin that is NOT peer-reviewed one. Also I have re-written majority of the report to make it a fit to the special issue of Forests.

Reviewer 3 Report

  1. In the introduction, more references are needed. Especially importance of harvesting residue, explanation of different harvesting methods, etc. 

  1. The hypothesis of this article should be clearly described in the introduction section.

  1. In the methods section, different harvesting sites number was not same. There was a large difference between cut-to-length and whole tree roadside case study numbers. Broadleaf and coniferous case study site characteristics should be in a separate table. Their quantity of harvesting residue would be different.  

  1. In the results section 3.2, some parts should be in the discussion section. 

  1. More exploration is needed in the discussion part according to the results

Author Response

1- New references on harvesting residues and methods were added to page 1 and page 2.

2- The hypothesis was added to page 4 as the reviewer asked.

3- I agree with reviewers and I have demonstrated the differences between pine and eucalypt in the manuscript. The observations were not the same and it is because there was less research conducted on WTH that CTL and we were really limited by this fact. I have explained this in the text of manuscript. 

4- Some parts of section 3.2 were moved to discussions as suggested.

5. More explanations were added to discussions in page 7 and 8 

Round 2

Reviewer 1 Report

The paper has been improved, no major revission are needed.

Nevertheless, some suggestions would improve It.

Figure 1: I suggest different colours for each harvesting system

Table 4 seems not to have been referenced un the text.

The table 4 should indicate the number of cases for each technology, and indicate the harvesting system.

Author Response

Thanks for providing further suggestions. Here is the points to address comments;

  • Figure 1 was changed as suggested and colours for each harvesting method were add to the figure in page 4.
  • Table 4 was referenced in page 6.
  • Harvesting method and number of cases were added to Table 4 in page 6.

Reviewer 2 Report

The paper is in accordance with my field of research and an interesting topic and I therefore accepted to review this manuscript. One of the main shortcomings of this study is the lack of scientific literature, I tried to suggest about fifteen papers in order to improve the paper.

Please find below all my comments

The introduction of the paper should include :

The definition of forest residues (Fine Woody Debris and Coarse woody debris) and their importance for the ecosystem functionning (soil fertility, biodiversity…etc.).

Current guidelines for sustainable biomass harvesting ?

Please find here some references to check and cite:

  1. Achat, D.L.; Deleuze, C.; Landmann, G.; Pousse, N.; Ranger, J.; Augusto, L. Quantifying consequences of removing harvesting residues on forest soils and tree growth—A meta-analysis. For. Ecol. Manag. 2015, 348, 124–141.
  2. Larrieu, L.; Cabanettes, A.; Gouix, N.; Burnel, L.; Bouget, C.; Deconchat, M. Post-harvesting dynamics of the deadwood profile: The case of lowland beech-oak coppice-with-standards set-aside stands in France. Eur. J. For. Res. 2019.
  3. Bessaad, Abdelwahab, Isabelle Bilger, and Nathalie Korboulewsky. "Assessing Biomass Removal and Woody Debris in Whole-Tree Harvesting System: Are the Recommended Levels of Residues Ensured?." Forests 12.6 (2021): 807.
  4. Titus, B.D.; Brown, K.; Helmisaari, H.-S.; Vanguelova, E.; Stupak, I.; Evans, A.; Clarke, N.; Guidi, C.; Bruckman, V.J.; Varnagiryte-Kabasinskiene, I. Sustainable forest biomass: A review of current residue harvesting guidelines. Energy Sustain. Soc. 2021, 11, 10.

Lines 36-37 : ‘’…while fine woody materials and coarse woody materials are left on the operation.’’ What is the diameter limit between the two classes ?

Lines 4-79 : The question is Why Intensive biomass harvesting or crown harvest can reduce soil fertility? The suitable answer is thet the most nutrient-rich compartements are located in the crown (leaves and fine wood < 7 cm).

Add more details please, check and cite studies about hervesting residues in European forests :

  1. Mälkönen, E. Effect of whole-tree harvesting on soil fertility. Silva Fenn. 1976, 10, 157–164.
  2. Achat, D.L.; Deleuze, C.; Landmann, G.; Pousse, N.; Ranger, J.; Augusto, L. Quantifying consequences of removing harvesting residues on forest soils and tree growth—A meta-analysis. For. Ecol. Manag. 2015, 348, 124–141.
  3. Kaarakka, L.; Tamminen, P.; Saarsalmi, A.; Kukkola, M.; Helmisaari, H.-S.; Burton, A.J. Effects of repeated whole-tree harvesting on soil properties and tree growth in a Norway spruce (Picea abies (L.) Karst.) stand. For. Ecol. Manag. 2014, 313, 180–187.

Line 91 : ‘’whole-tree harvesting’’ instead of ‘’whole-tree to the roadside’’

Lines 108 : Title ?

Line 116 : distance between each transect ? How is the distibution of the 50 samples ? please specify

Line 261 : cite here the results from this new published study about hervesting residues in European forests : Bessaad, Abdelwahab, Isabelle Bilger, and Nathalie Korboulewsky. "Assessing Biomass Removal and Woody Debris in Whole-Tree Harvesting System: Are the Recommended Levels of Residues Ensured?." Forests 12.6 (2021): 807.

Line 289 : I agree that ‘’High share of needles may positively contribute to maintaining nutrients…‘’. Yes Needles and leaves are the most nutrien-rich compartement compared to other compartements but I’m not sure that only 20-30% needles would maintain soil fertility ! More than 2/3 is exported !

Please check and cite:

  1. Bessaad, Abdelwahab, and Nathalie Korboulewsky. "How much does leaf leaching matter during the pre-drying period in a whole-tree harvesting system?." Forest Ecology and Management 477 (2020): 118492.

Lines 294 : What are these impacts ? please give the readers more details about impacts on soil, water, biodiversity…etc.

See and cite :

  1. Kimmins, J.P., 1976. Evaluation of the consequences for future tree productivity of the loss of nutrients in whole-tree harvesting. For. Ecol. Manage. 1, 169–183
  2. Johnson, D.W., Trettin, C.C., Todd, D.E., 2016. Changes in forest floor and soil nutrients in a mixed oak forest 33 years after stem only and whole-tree harvest. For. Ecol. Manage. 361, 56–68
  3. Thiffault, E., Hannam, K.D., Pare, D., Titus, B.D., Hazlett, P.W., Maynard, D.G., Brais, S., 2011. Effects of forest biomass harvesting on soil productivity in boreal and tempe-rate forests – a review. Environ. Rev. 19, 278–30
  4. Berch, S.; Morris, D.; Malcolm, J. Intensive forest biomass harvesting and biodiversity in Canada: A summary of relevant issues.For. Chron. 2011, 87, 478–487
  5. Ranius, T.; Hämäläinen, A.; Egnell, G.; Olsson, B.; Eklöf, K.; Stendahl, J.; Rudolphi, J.; Sténs, A.; Felton, A. The effects of logging residue extraction for energy on ecosystem services and biodiversity: A synthesis. J. Environ. Manag. 2018, 209, 409–425.
  6. Miettinen, J.; Ollikainen, M.; Nieminen, T.M.; Ukonmaanaho, L.; Laurén, A.; Hynynen, J.; Lehtonen, M.; Valsta, L. Whole-tree harvesting with stump removal versus stem-only harvesting in peatlands when water quality, biodiversity conservation and climate change mitigation matter. For. Policy Econ. 2014, 47, 25–35.

Line 298 : Conclusion : The conclusion usually does not contain a reference in most journals, there is no need to use citations in the "Conclusion" part of your article !

You should rewrite the conclusion section because it looksmore like a discussion section !!

Please :

  1. Restate the research problem addressed in the paper.
  2. Summarize the main points (Do not simply summarize the points already made in the body).
  3. State the significance or results (interpret your findings at a higher level of abstraction).
  4. Show what your findings mean to forest managers and readers..
  5.  

Lines 298 – 300 : Conclusion should not include figures citations or statistical methods, please rephrase these sentence ‘’The weight of harvesting residues per ha varied highly across different operational sites (Figure 1). From the statistical test the weight of harvesting residues per ha significantly depended on the applied harvesting method’’

Line 306 : How much % ?

Author Response

Thanks for providing suggestion on the paper. Here is the list of revisions to address your feedback;

  • Definition of forest residues and their importance were added in page 1.
  • Guidelines and suggested references were added in page 2.
  • The diameter limit between two classes were added in page 1.
  • The explanations on why intensive biomass recovery leads into reduce soil fertility and suggested references were all added in page 2.
  • Whole tree harvesting was replaced with whole tree to road side in page 3.
  • Title of method was added in page 3.
  • Information on distance between transects and distribution were added in page 3.
  • New information and suggested reference were added on recommended level of harvesting residues in page 8.
  • New information and suggested reference were added about share of needles and contributions to soil health in page 9.
  • The impacts of biomass recovery on soil, water and biodiversity were added using suggested references in page 9.
  • Conclusions was changed and revised as suggested by reviewer. The research problem was restated, main point were written, significance of results and benefits to industry were added in page 9.
  • Percentage of needles share were added. 
  • The section for future required research were retained plus its references. I understand and appreciate the reviewer point on citing in conclusion but these points are very important for readers and have been borrowed from references so they were kept with not much change.

Reviewer 3 Report

The article titled ‘Analyzing the impact of harvesting methods on the quantity of harvesting residues: An Australian case study’ is an interesting paper. It can be published.

Author Response

Thank you for providing useful feedback. Much appreciated.